# Association between Health-Related Physical Fitness and Self-Rated Risk of Depression in Adolescents: Dados Study

**DOI:** 10.3390/ijerph17124316

**Published:** 2020-06-17

**Authors:** Carlos Bou-Sospedra, Mireia Adelantado-Renau, Maria Reyes Beltran-Valls, Diego Moliner-Urdiales

**Affiliations:** LIFE Research Group, Universtiy Jaume I, PC 12071 Castellón de la Plana, Spain; al379860@uji.es (C.B.-S.); adelantm@uji.es (M.A.-R.); vallsm@uji.es (M.R.B.-V.)

**Keywords:** cardiorespiratory fitness, weight status, body mass index, psychological well-being, depression, adolescence

## Abstract

Depression is the most common mental disorder, affecting around 5% of adolescents. Physical fitness is considered a powerful marker of physical and mental health. The scientific results on the relationship between physical fitness and depression in the adolescent population are mixed. Therefore, the aim of the current study was to analyse the association between objectively assessed physical fitness and self-rated risk of depression in a group of adolescents. A total of 225 participants (44% girls), aged 13.9 ± 0.3 years, from the Deporte, ADOlescencia y Salud (DADOS) study were included in the analyses. Field-based Assessing Levels of Physical fitness and Health in Adolescents (ALPHA) health-related fitness test battery was used to objectively assess physical fitness components. The Behavior Assessment System for Children (BASC) level 3 was used to evaluate self-rated risk of depression. Our results showed that self-rated risk of depression was inversely associated with cardiorespiratory fitness (*β* = −0.172), as well as positively associated with body mass index (*β* = 0.146) and waist circumference (*β* = 0.137) (all *p* < 0.05). Adolescents with low levels of cardiorespiratory fitness had significantly higher odds of self-rated risk of depression (OR = 7.17; 95% CI, 1.51–33.95). These findings suggest that health-related physical fitness, particularly cardiorespiratory fitness and body composition, is associated with depression in adolescents.

## 1. Introduction

Depression is the most common mental disorder, and it could be defined as low mood, pessimism, and apathy experienced over an extended length of time [1]. Adolescent depression has around 5% of prevalence and a tendency to increased occurrence over recent years has been observed [2]. Adolescence and early adulthood are critical time frames in terms of depression onset, probably due to a confluence of physical, hormonal, social, emotional, neuronal, behavioural psychological and sleep changes [3,4,5]. Depression during adolescence appears to be associated with some unhealthy behaviours, such as low levels of physical activity [6,7], and with some unhealthy attributes, such as obesity [8]. Hence, adolescence represents a critical period of depression risk that underscores the need to identify factors that can prevent or minimize the deleterious effects of this mental disease [9].

Health-related physical fitness is as a set of attributes related to a person’s ability to perform physical activity, which includes cardiorespiratory fitness, musculoskeletal and motor capacities, as well as body composition [10]. It is determined by a combination of genetic inheritance and health-related behaviours [11,12], and it is considered a powerful marker of physical and mental health [13]. Indeed, findings from cross-sectional [14,15] and longitudinal studies [16] have shown an inverse association between cardiorespiratory fitness and depression in adolescents. With regard to body composition, a meta-analysis reported that obese adolescents had a 40% greater risk of being depressed [8]. Although muscular fitness and speed-agility have shown to be associated with depression in children [17,18] and young adults [19], to our knowledge, no prior studies have investigated these associations in adolescent population.

Expanding the previous scientific literature with an overall approach to the relationship between all health-related physical fitness components and risk of depression is needed. Increasing knowledge about the influence of low health-related physical fitness on risk of depression in adolescents would help to stablish future promotion strategies aimed to improve adolescents’ mental health. Based on the previous literature about health-related physical fitness and depression in adolescents [14,15,16] and other populations [17,18,19], we hypothesized that adolescents with lower levels of physical fitness are at higher risk of depression. Therefore, the aim of the current study was to explore the associations between all health-related physical fitness components and self-rated risk of depression in a sample of adolescents.

## 2. Materials and Methods

### 2.1. Study Design and Sample Selection

This study is part of the Deporte, ADOlescencia y Salud (DADOS) research project, a 3-year longitudinal study aimed to analyse the influence of physical activity on health, academic performance, and psychological wellness through adolescence. All participants were recruited from secondary schools located in Castellon (Spain), and met the general inclusion criteria: enrolled in second grade of secondary school, and without diagnosed physical or neurological chronic diseases (as reported by participants’ parents). The results presented in this study belong to baseline data obtained between February and May of 2015. A total of 225 adolescents aged 13.9 ± 0.3 years (44% girls) completed the baseline assessment with valid data for health-related physical fitness components and self-rated risk of depression.

Adolescents and their parents or guardians were informed about the nature and characteristics of the study, and all provided a written informed consent. The DADOS study protocol was designed in accordance with the ethical guidelines of the Declaration of Helsinki 1964 (last revision of Fortaleza, Brazil, 2013), and was approved by the Research Ethics Committee of the University Jaume I of Castellon (Spain).

### 2.2. Health-Related Physical Fitness

The Assessing Levels of Physical fitness and Health in Adolescents (ALPHA) health-related fitness field-based test battery was used, in order to objectively assess physical fitness components [10].

Cardiorespiratory fitness was assessed by the 20 m shuttle run test [20]. Briefly, each participant runs straight between 2 lines 20 m apart, while keeping the pace with audio signals. The test was completed when participants could not reach the end lines at the pace of the audio signals 2 consecutive times, or when they stopped because of fatigue. The number of laps (20 m each) was registered and used in the analyses. Cardiorespiratory fitness level was dichotomized into low and high according to the 60th sex-specific percentile based on normative values [21].

Upper limb muscular strength was measured with a hand dynamometer with adjustable grip (TKK 5101 Grip D; Takey, Tokyo, Japan). Briefly, the participant squeezes gradually and continuously for at least 2 s, performing the test with the right and left hands in turn, and using the optimal grip span. The maximum score in kilograms for each hand was recorded, and the average of the scores achieved in both handgrip tests was used in the analyses. The upper limb muscular strength level was dichotomized into low and high, according to the 60th sex-specific percentile, based on normative values [21].

Lower limb muscular strength was assessed with the standing broad jump test. Briefly, the participant jumps as far as possible from a starting position immediately behind a line, standing with feet approximately shoulder’s width apart. The maximum score in centimetres was used in the analyses. Lower limb muscular strength level was dichotomized into low and high, according to the 60th sex-specific percentile based on normative values [21].

Speed–agility was assessed with the 4 × 10 m shuttle run test. Briefly, the participant runs as fast as possible 4 times between two parallel lines 10 m apart. The minimum time taken to complete the test was used in the analyses. For analytic purposes, this variable was multiplied by −1, so a higher score indicates better speed-agility. Speed-agility level was dichotomized into high and low, according to the 66th sex-specific percentile.

Weight was measured with an electronic scale (model 861; Seca, Hamburg, Germany) to the nearest 0.1 kg. Height was measured in the Frankfort plane with a wall-mounted stadiometer (model 213; Seca, Hamburg, Germany) to the nearest 0.1 cm. Weight and height were measured in duplicate, and the average was used for data analyses. Body mass index (BMI) was calculated as weight in kilograms divided by the square of the height in meters (kg/m^2^). Participants were classified into normal weight or overweight/obese, according to the international age and sex-specific BMI cut-offs proposed by Cole et al. [22].

Waist circumference was measured to the nearest 1 mm with a non-elastic tape (Harpenden anthropometric tape; Holtain Ltd., Crymych, UK) applied horizontally midway between the lowest rib margin and the iliac crest, at the end of gentle expiration with the participant in a standing position. Speed-agility level was dichotomized into high and low according to the 33rd sex-specific percentile.

Tricipital and subscapular skinfold thicknesses were measured by duplicate to the nearest 0.2 mm with a Holtain Caliper (Crymmych, UK). Body fat percentage was calculated by the equation described by Slaughter, Lohman, and Boileau [23].

### 2.3. Risk of Depression

The Behavior Assessment System for Children (BASC), level 3 for adolescents, which has shown extensive psychometric properties in both non-referred and clinical populations with reliabilities for the subscales ranging from 0.80 to 0.87, Spanish version [24] was completed by participants to assess self-rated risk of depression [25]. For risk of depression, standard T-score with an average of 50 and a standard deviation of 10 points were used in the analyses. Risk of depression was dichotomized into “non-risk” (<60) and “at risk” (≥60), according to the established cut off point [25].

### 2.4. Covariates

According to the previous scientific literature, all the analyses were adjusted by pubertal stage and socioeconomic status, due its strong association with physical fitness components [26,27,28,29] and depression [30,31,32,33] in the adolescent population.

Pubertal stage was self-reported according to the five stages described by Tanner and Whitehouse [34]. It is based on external primary and secondary sexual characteristics, which are described by the participants using standardized pictures.

Socioeconomic status was reported by The Family Affluence Scale (FAS) questionnaire, developed by Currie et al. [35]. It was used as a proxy of socioeconomic status, which is based on material conditions in the family, such as car ownership, bedroom occupancy, computer ownership, and home internet access. Results ranging from 0 to 8 points were categorized as follows: low 1–2, medium 3–5, high 6–8.

### 2.5. Statistical Analyses

Study sample characteristics are presented as mean ± standard deviation (SD) and percentages for continuous and categorical variables, respectively. As preliminary analyses did not show a significant interaction of sex with the study variables in relation to risk of depression (*p* > 0.01), all analyses were performed for the whole sample.

Linear regression analyses were conducted to analyses the association between self-rated risk of depression with each health-related physical fitness components. Each physical fitness components at baseline was entered as the independent variable and self-rated depression was entered as depended variable in separate models. To avoid the influence of sex, pubertal stage, and socioeconomic status, all models included these factors as covariates. The Durbin-Watson coefficient, ranging from 0 to 4, was reported to assess the autocorrelation of the residuals. Binary logistic regression models were used to assess the likelihood of being at risk of depression depending on the level of each physical fitness component (low vs. high), after adjusting for sex, pubertal stage, and socioeconomic status. In these analyses the R^2^ of Cox/Snell is reported. All the analyses were performed using the IBM SPSS Statistics for Windows version 22.0 (Armonk, NY, USA: IBM Corp), and the level of significance was set to *p* < 0.05.

## 3. Results

Descriptive characteristics of the study population by sex are presented at Table 1.

Linear regression analyses were performed to analyse the association between each health-related physical fitness component and self-rated risk of depression adjusting for sex, pubertal stage and socioeconomic status (Table 2). Cardiorespiratory fitness, but not muscular strength and speed agility, was inversely associated with self-rated risk of depression (*β* = −0.172, *p* < 0.05). BMI and waist circumference were positively associated with self-rated risk of depression (*β* = 0.146, *p* < 0.05; *β* = 0.137, *p* < 0.05). Similar results were found in partial correlations analyses (Appendix A).

Binary logistic regression analyses examining the associations between categories for each health-related physical fitness component and categories of self-rated risk of depression are shown in Table 3. According to health-related physical fitness components, adolescents with low levels of cardiorespiratory fitness showed higher likelihood to be at self-rated risk of depression (OR = 7.17; 95% CI, 1.51–33.95). No statistically significant associations between muscular strength or speed-agility categories and self-rated risk of depression were identified.

## 4. Discussion

To our knowledge, this is the first study analysing the association between a complete range of health-related physical fitness components and self-rated risk of depression in adolescents. The main findings of the present study revealed that cardiorespiratory fitness was inversely, but weakly, associated with self-rated risk of depression. Moreover, BMI and waist circumference were positively, but weakly, associated with self-rated risk of depression in adolescents. Additionally, adolescents with low levels of cardiorespiratory fitness showed higher likelihood to be at risk of depression. Our results complement the prior literature, indicating a unique contribution of each health-related fitness component on the risk of depression in adolescents.

Part of our data are in accordance with the results reported by Rieck et al. [14], suggesting that adolescents with low levels of cardiorespiratory fitness have an increased likelihood of depressive symptoms. In addition, our results partially concur with previous cross-sectional [15] and longitudinal [16] studies, showing an inverse association between cardiorespiratory fitness and levels of depression in girls but not in boys. We speculate that the association found in this study is related to the physiological effects of regular physical activity [36,37], since cardiorespiratory fitness has been shown to improve through sufficient physical activity practice in adolescents [38]. Although the mechanisms explaining the association between mental health and physical activity remain to be elucidated [36,37,38,39], it is likely that improvements in cardiorespiratory fitness through physical activity may lead to lower risk of depression in adolescents.

With regard to muscular strength and speed-agility, no association was found with self-rated risk of depression in our sample. This is the first study focused on adolescents, directly hampering comparisons among studies. Our results concur with the previous data reported in children showing null associations [17,18]. However, the results in young adults [19] have shown an inverse association between muscular fitness and depressive symptoms. The lack of agreement among studies might be due to the fact that adolescence is a critical period of life, in which the confluence of physical and hormonal changes could lead to different growth curves [40].

According to body composition, our results revealed positive, but weak, associations of BMI and waist circumference with self-rated risk of depression. These findings concur with previous studies, revealing a positive association between BMI and risk of depression in adolescents [41]. However, the association between body composition and risk of depression is controversial, since Rieck et al. [14] reported a null association. The findings of the present study could be partially explained by the fact that a healthy body composition has been related to improved body satisfaction, self-esteem, and social functioning, which in turn, may lead to lower depressive symptoms [42].

Our data may have significant implications from a public health point of view for promoting psychological well-being in adolescents. Indeed, high levels of cardiorespiratory fitness and a healthy composition seem to be protective factors for risk of depression. Active behaviours should be promoted among adolescents by educators, families, and policy makers, in order to improve cardiorespiratory fitness and body composition levels. This is especially important in this age group, since adolescence is a crucial period of life, during which health-related behaviours are established [43].

### Limitations and Strengths of the Study

The results of this study should be interpreted with caution, because the cross-sectional design does not allow us to infer causality between health-related fitness components and self-rated risk of depression. The Spanish version of the FAS questionnaire was used as proxy measure of socioeconomic status, but no specific validation study is available for Spanish adolescents. The strengths of this study comprised the relatively large and homogeneous sample in terms of age and sex, the use of validated tests to assess health-related physical fitness components and self-rated risk of depression, and the inclusion of several confounders in the statistical analyses with great relevance for physical fitness and depression (i.e., pubertal development and socioeconomic status).

## 5. Conclusions

In conclusion, our findings add new information about the relationship between objectively health-related physical fitness components and self-rated risk of depression in adolescents. Specifically, our results indicated an association of cardiorespiratory fitness and body composition with self-rated risk of depression. Nevertheless, muscular strength and speed agility do not seem to be associated with a self-rated risk of depression. These attributes should be considered in order to reduce the risk of depression in adolescents. Future longitudinal and interventional studies are needed, in order to identify the mechanisms involved in the associations between health-related physical fitness components and psychological wellbeing in adolescents.

## Figures and Tables

**Table 1 ijerph-17-04316-t001:** Descriptive characteristics of the study sample by sex.

Variables	All (n = 225)	Boys (n = 127)	Girls (n = 98)
Age (y)	13.89 ± 0.29	13.89 ± 0.29	13.89 ± 0.29
Pubertal stage II-V (%)	8.0/34.7/46.7/10.7	10.2/33.1/42.5/14.2	5.1/36.7/52.0/6.1
Socioeconomic status (0–8)	4.12 ± 1.42	4.29 ± 1.51	3.98 ± 1.33
Health-related physical Fitness			
Cardiorespiratory fitness (laps)	68.43 ± 23.37	78.85 ± 20.34	54.93 ± 19.95
Upper limb muscular strength (kg)	29.31 ± 6.06	30.93 ± 6.81	27.20 ± 4.09
Lower limb muscular strength (cm)	174.52 ± 24.66	180.94 ± 23.55	166.19 ± 23.65
Speed-Agility (s)	12.37 ± 0.88	12.02 ± 0.69	12.88 ± 0.85
Body composition			
Height (cm)	163.14 ± 8.00	164.69 ± 8.65	161.13 ± 6.60
Weight (kg)	53.93 ± 8.89	54.07 ± 8.92	53.74 ± 8.91
Body mass index (kg/m^2^)	20.19 ± 2.56	19.83 ± 2.23	20.65 ± 2.89
Waist circumference (cm)	67.19 ± 5.52	67.81 ± 5.03	66.38 ± 6.03
Total body fat (%)	9.69 ± 3.32	18.30 ± 6.83	25.20 ± 5.53
Self-rated risk of depression			
Score	45.78 ± 7.75	44.84 ± 5.71	47.01 ± 9.69
At risk (%)	8 (3.6)	2 (1.6)	6 (6.1)

Data are presented as mean ± standard deviation or percentages.

**Table 2 ijerph-17-04316-t002:** Linear regression analyses of health-related physical fitness components with self-rated risk of depression.

Variables	R^2^	B	95% CI		*β*	*P*	DW
			Lower	Upper			
Cardiorespiratory fitness	0.049	−0.057	−0.107	−0.007	−0.172	0.025	1.99
Upper limb muscular strength	0.027	0.006	−0.187	0.199	0.005	0.953	1.98
Lower limb muscular strength	0.027	−0.004	−0.048	0.040	−0.013	0.856	1.98
Speed-Agility *	0.034	−0.869	−2.213	0.474	−0.099	0.204	2.00
Body mass index	0.046	0.442	0.028	0.857	0.146	0.037	2.02
Waist circumference	0.044	0.192	0.001	0.384	0.137	0.049	2.03
Total body fat	0.038	0.133	−0.028	0.294	0.123	0.104	2.00

Analyses adjusted for sex, pubertal stage, and socioeconomic status. Statistically significant values are highlighted in bold. B, unstandardized coefficient; *β*, standardized coefficient; CI, confidence interval; DW, Durbin-Watson coefficient. * For analytic purposes, speed-agility variable was multiplied by −1, so a higher score indicates better speed-agility.

**Table 3 ijerph-17-04316-t003:** Logistic regression analyses of health-related physical fitness components categories with self-rated risk of depression.

Variables		n (%)	R^2^	OR	95% CI
Cardiorespiratory fitness	High	188 (83.56)		1	Reference
	Low	37 (16.44)	0.063	**7.17**	**1.51**–**33.95**
Upper limb muscular strength	High	125 (55.6)		1	Reference
	Low	100 (44.4)	0.037	1.14	0.25–5.19
Lower limb muscular strength	High	141 (62.7)		1	Reference
	Low	84 (37.3)	0.038	0.75	0.14–3.96
Speed-Agility	High	75 (33.3)		1	Reference
	Low	150 (66.7)	0.038	0.75	0.14–4.06
Body mass index	Non-overweight	200 (88.9)		1	Reference
	Overweight	25 (11.1)	0.038	1.62	0.28–9.31
Waist circumference	Low	152 (67.6)		1	Reference
	High	73 (32.4)	0.040	2.37	0.27–20.79
Total body Fat	Low	145 (64.4)		1	Reference
	High	80 (35.6)	0.043	3.19	0.37–27.45

Analyses were adjusted for sex, pubertal stage, and socioeconomic status. Statistically significant values are in bold. OR, Odds Ratio; CI, confidence interval.

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
