# Peer review of "Association between Health-Related Physical Fitness and Self-Rated Risk of Depression in Adolescents: Dados Study"

_ijerph, 2020, doi:10.3390/ijerph17124316_

Round 1

Reviewer 1 Report

The revised version has been improved

Author Response

Thank you for your comment and your positive report.

Reviewer 2 Report

Dear Authors,

We agree totally with your article that is clear, well justified and well elaborated. The only concern we have is The Family Affluence Scale questionnaire you did use. Is this instrument validated for Spanish language, if yes please indicate the authors, if not please indicate what you did for the translation, …  and note this as a limit of your study.  Congratulations and wishes of success for your article that we will recommend for publication.

Author Response

Dear Prof. Dr. Paul B. Tchounwou,

My co-authors and I are pleased to respond the reviewer and academic editor comments, and resubmit a revised version of the paper entitled: “Association between health-related physical fitness and risk of depression in healthy adolescents: DADOS study” submitted for the special issue “Fitness, Physical Activity, and Health in Youth” at International Journal of Environmental Research and Public Health (No. IJERPH-778764).

Yours sincerely,

The authors

#REVIEWER 2

Comment

Dear Authors,

We agree totally with your article that is clear, well justified and well elaborated. The only concern we have is The Family Affluence Scale questionnaire you did use. Is this instrument validated for Spanish language, if yes please indicate the authors, if not please indicate what you did for the translation, …  and note this as a limit of your study.  Congratulations and wishes of success for your article that we will recommend for publication.

Response

Thank you for the valuable comment. The Family Affluence Scale (FAS) questionnaire is a valid proxy of socioeconomic status in young people and has been widely used in large epidemiological studies such as the Health Behaviour in School-aged Children (HBSC). HBSC study, with the support of the World Health Organization, is analyzing the social determinants of health, social media, mental health, and other aspects of adolescent well-being every four years in 50 countries across Europe (including Spain) and North America (more info at http://www.hbsc.org/index.aspx).

In our study, we use the Spanish version of FAS questionnaire reported by Currie et al., (2008) and provided by the consortium of the HBSC. In our best of knowledge, there is no validation study about the Spanish version of FAS questionnaire. According with the reviewer suggestion, this issue has been included in the limitations of the study (Please, see lines 221-222).

#ACADEMIC EDITOR

Comment

Dear authors, 

Thank you for your revised version of your manuscript “Association between health-related physical fitness and risk of depression in healthy adolescents: DADOS study”. The authors are commended for a responsive revision in which all major comments have been addressed. We are therefore happy to accept the manuscript. 

On a minor note, please double check in the proof of the article whether it sound be 60th percentile for the categorization of speed-agility (it says 66th).

King regards,

Pontus

Response

Thank you for your appreciation. In our study, physical fitness components were assessed through the ALPHA (Assessing Levels of Physical fitness and Health in Adolescents) health-related fitness field-based test battery. Each physical fitness component was categorized according the European normative values reported by Tomkinson et al., (2018). However, the speed-agility test used by Tomkinson et al., (2018) is based on a different protocol than ours. For this reason, and based on previous categorization analysis performed by our research group, speed-agility values were divided into tertiles, and third tertile (percentile 66th) was categorized as “high speed-agility level”.

References

Tomkinson GR, Carver KD, Atkinson F, et al. European normative values for physical fitness in children and adolescents aged 9-17 years: Results from 2 779 165 Eurofit performances representing 30 countries. Br J Sports Med. 2018;52(22):1445-1456.

Currie C, Molcho M, Boyce W, Holstein B, Torsheim T, Richter M. Researching health inequalities in adolescents: The development of the Health Behaviour in School-Aged Children (HBSC) Family Affluence Scale. Soc Sci Med. 2008;66(6):14

This manuscript is a resubmission of an earlier submission. The following is a list of the peer review reports and author responses from that submission.

Round 1

Reviewer 1 Report

The research is very valuable. The results and the conclusions of the study are also interesting and valuable. However authors should expand the discussion, trying to explain the reasons for the relationships found, although this is not the objective of the work. I suggest to describe even more precisely last paragraph in the discussion in the form of a separate chapter entitled "Limitation of the study"

Author Response

My co-authors and I are pleased to respond the reviewers’ comments and resubmit a revised version of the paper entitled: “Association between health-related physical fitness and self-rated risk of depression in adolescents: DADOS study” submitted for the special issue “Fitness, Physical Activity, and Health in Youth” at International Journal of Environmental Research and Public Health (No. IJERPH-778764). We have explained the points raised by the reviewers and we have modified the manuscript as requested. The changes made on the original manuscript appear in red text for ease of identification.

Yours sincerely,

The authors

REVIEWER 1

Comment 1

The research is very valuable. The results and the conclusions of the study are also interesting and valuable.

Response

Thank you for the comment.

Comment 2

However, authors should expand the discussion, trying to explain the reasons for the relationships found, although this is not the objective of the work.

Response

We thank the reviewer very much for the suggestion. We have reformulated and expanded the discussion section. Please, see lines 186 to 240.

Comment 3

I suggest to describe even more precisely last paragraph in the discussion in the form of a separate chapter entitled "Limitation of the study"

Response

According to the reviewer’s comment, the section “Limitations and strengths of the study” has been expanded and included as an independent section. Please, see lines 225 to 232.

Reviewer 2 Report

The authors investigated the associations between health-related physical fitness and symptoms of depression among a sample of 225 participants in the stage of early adolescence. The authors observed that higher scores of objectively assessed cardiovascular fitness were associated with lower scores of self-rated symptoms of depression. Similarly, higher BMI and waist circumference indices were associated with a higher risk of higher self-reported symptoms of depression. The authors conclude that lower cardiovascular fitness indices are associated with higher risks of self-reported symptoms of depression.

Abstract: “The association between fitness and depression in healthy adolescents is poorly understood”; I suggest to attenuating this statement, as there is already sufficient evidence of that kind of association. Report the gender-ratio. Spell-out DADOS, when mentioning the acronym for the first time. I suggest to specifying the wording such as ‘self-rated symptoms of depression’, ‘objectively assessed fitness indices’, ‘higher cardiorespiratory fitness indices were inversely associated with a higher risk of self-rated symptoms of depression’, and similar. As regards the following beta-weight: β = -0.063; p < 0.05, please note that despite the ‘significant’ p-value, objectively assessed fitness indices were able to explain about 6% of the variance of self-rated symptoms of depression, or simply put: the predictive power of objectively assessed fitness indices is spurious.

Introduction: delete Beck et al 1961, as this reference describes a tool, but not diagnostic criteria. Rather cite the DSM-5 (American Psychiatric Association, 2013).

“probably due to a confluence of physical, hormonal, social, emotional and psychological changes”. Please add also neuronal changes (Giedd et al., 1999; Gogtay et al., 2004; Paus, Keshavan, & Giedd, 2008; Paus et al., 1999; Spear, 2000) and changes in sleep (Brand et al., 2014; Brand et al., 2017); further, as regards stability and changes in physical activity indices, the authors should give a closer look at (Madsen, McCulloch, & Crawford, 2009); as regards the protective power of regular physical activity for symptoms of depression and parent-child issues, please give a close look at (Sigfusdottir, Asgeirsdottir, Sigurdsson, & Gudjonsson, 2011). For the emergence and maintenance of symptoms of depression in females, see (Bor, Dean, Najman, & Hayatbakhsh, 2014; Hyde, Mezulis, & Abramson, 2008). Next, the association between mental health and physical activity is not that linear and easy as suspected: (Opdal et al., 2019) nicely showed that changes and stability in psychological functioning and changes and stability in physical activity indices were completely unrelated. Next, the authors must give a very close look at (Bailey, Hetrick, Rosenbaum, Purcell, & Parker, 2017), when setting-up the theoretical background of their study.

“Prior research has shown a strong positive association between weight status (overweight or obese adolescents) and depression [22,23]. Findings from reviews and meta-analyses studies have shown a positive relationship between weight status and depression [24]. In addittion (sic!), a systematic review and meta-analyses of longitudinal studies [7] suggested a bi-directional association between depression and obesity in adolescents.”; here, it is not clear, why this paragraph has been introduced; if weight is a dependent or an independent variable, then the title must be changed. Likewise, when weight is an issue, weight gain is a proxy of symptoms of eating disorders, and as such, the authors do not assess a “non-pathological” sample anymore.

Overall, the Introduction section needs a thorough revision; key concepts must be introduced and described in a much more detailed fashion; the state-of-the-art must be thoroughly reported. In a similar vein, I strongly suggest to formulating hypotheses: based on the most recent findings (see above all see the more recent references mentioned above), what kind of pattern of results do they expect from their data and why? Likewise, do the present results add to the current literature in an important way, and if so, to what extent. Last, it remains puzzling, if among a sample of psychologically healthy adolescents (as stated by the authors) one could expect more than mere floor effects of self-rated symptoms of depression.

Methods: How did the authors know or assess that participants were “free of any chronic disease.” Here, the reader needs much more information. Relatedly, does “chronic disease” relate exclusively to physical/somatic issues such as diabetes, obesity, sleep-disordered breathing such as Obstructive Sleep Apnea, neurological issues such as early-onset of multiple sclerosis, asthma, celiac disease, or irritable bowel disease, just to name a few, or does “chronic disease” also relate to psychiatric diseases such as ADHD (prevalence rate: 5.6% (C. A. Polanczyk, de Lima, Horta, Biederman, & Rohde, 2007; C. A. Polanczyk, Willcutt, Salum, Kieling, & Rohde, 2014; G. V. Polanczyk, Salum, Sugaya, Caye, & Rohde, 2015), or internalizing problems (Bor et al., 2014)?

The Declaration of Helsinki was first stated in 1964, not 1961.

Risk of depression; please specify, who completed which part of the Behaviour Assessment System for Children.

Statistical analysis: I strongly recommend to report effect sizes; this holds particularly true, as the sample is quite large; it follows that due to the sample size the p-value might get ‘significant’, while mean differences are still spurious and trivial.

Results: Table 1; please report all statistical indices; reporting just p-values does not make sense (Wasserstein, Schirm, & Lazar, 2019; Zhu, 2012, 2016).

Discussion: as for the Introduction section, the Discussion section needs a thorough revision. Among others and specifically, the authors should critically comment on the low beta-weights.

References

American Psychiatric Association. (2013). Diagnostic and Statistical Manual of Mental Disorders 5th edition: DSM 5. Arlington VA: American Psychiatric Association.

Bailey, A. P., Hetrick, S. E., Rosenbaum, S., Purcell, R., & Parker, A. G. (2017). Treating depression with physical activity in adolescents and young adults: a systematic review and meta-analysis of randomised controlled trials. Psychol Med, 1-20. doi:10.1017/s0033291717002653

Bor, W., Dean, A. J., Najman, J., & Hayatbakhsh, R. (2014). Are child and adolescent mental health problems increasing in the 21st century? A systematic review. Aust N Z J Psychiatry, 48(7), 606-616. doi:10.1177/0004867414533834

Brand, S., Kalak, N., Gerber, M., Clough, P. J., Lemola, S., Puhse, U., & Holsboer-Trachsler, E. (2014). During early and mid-adolescence, greater mental toughness is related to increased sleep quality and quality of life. J Health Psychol. doi:10.1177/1359105314542816

Brand, S., Kalak, N., Gerber, M., Clough, P. J., Lemola, S., Sadeghi Bahmani, D., . . . Holsboer-Trachsler, E. (2017). During early to mid adolescence, moderate to vigorous physical activity is associated with restoring sleep, psychological functioning, mental toughness and male gender. J Sports Sci, 35(5), 426-434. doi:10.1080/02640414.2016.1167936

Giedd, J. N., Blumenthal, J., Jeffries, N. O., Castellanos, F. X., Liu, H., Zijdenbos, A., . . . Rapoport, J. L. (1999). Brain development during childhood and adolescence: a longitudinal MRI study. Nat Neurosci, 2(10), 861-863. doi:10.1038/13158

Gogtay, N., Giedd, J. N., Lusk, L., Hayashi, K. M., Greenstein, D., Vaituzis, A. C., . . . Thompson, P. M. (2004). Dynamic mapping of human cortical development during childhood through early adulthood. Proc Natl Acad Sci U S A, 101(21), 8174-8179. doi:10.1073/pnas.0402680101

Hyde, J. S., Mezulis, A. H., & Abramson, L. Y. (2008). The ABCs of depression: integrating affective, biological, and cognitive models to explain the emergence of the gender difference in depression. Psychol Rev, 115(2), 291-313. doi:10.1037/0033-295X.115.2.291

Madsen, K. A., McCulloch, C. E., & Crawford, P. B. (2009). Parent modeling: perceptions of parents' physical activity predict girls' activity throughout adolescence. J Pediatr, 154(2), 278-283. doi:10.1016/j.jpeds.2008.07.044

Opdal, I. M., Morseth, B., Handegård, B. H., Lillevoll, K., Ask, H., Nielsen, C. S., . . . Rognmo, K. (2019). Change in physical activity is not associated with change in mental distress among adolescents: the Tromsø study: Fit Futures. BMC Public Health, 19(1), 916-916. doi:10.1186/s12889-019-7271-6

Paus, T., Keshavan, M., & Giedd, J. N. (2008). Why do many psychiatric disorders emerge during adolescence? Nat Rev Neurosci, 9(12), 947-957. doi:10.1038/nrn2513

Paus, T., Zijdenbos, A., Worsley, K., Collins, D. L., Blumenthal, J., Giedd, J. N., . . . Evans, A. C. (1999). Structural maturation of neural pathways in children and adolescents: in vivo study. Science, 283(5409), 1908-1911. Retrieved from http://www.ncbi.nlm.nih.gov/pubmed/10082463

Polanczyk, C. A., de Lima, M. S., Horta, B. L., Biederman, J., & Rohde, L. A. (2007). The worldwide prevalence of ADHD: a systematic review and metaregression analysis. Am J Psychiatry, 164(6), 942-948. doi:10.1176/ajp.2007.164.6.942

Polanczyk, C. A., Willcutt, E. G., Salum, G. A., Kieling, C., & Rohde, L. A. (2014). ADHD prevalence estimates across three decades: an updated systematic review and meta-regression analysis. Int J Epidemiol, 43(2), 434-442. doi:10.1093/ije/dyt261

Polanczyk, G. V., Salum, G. A., Sugaya, L. S., Caye, A., & Rohde, L. A. (2015). Annual research review: A meta-analysis of the worldwide prevalence of mental disorders in children and adolescents. J Child Psychol Psychiatry, 56(3), 345-365. doi:10.1111/jcpp.12381

Sigfusdottir, I. D., Asgeirsdottir, B. B., Sigurdsson, J. F., & Gudjonsson, G. H. (2011). Physical activity buffers the effects of family conflict on depressed mood: a study on adolescent girls and boys. J Adolesc, 34(5), 895-902. doi:10.1016/j.adolescence.2011.01.003

Spear, L. P. (2000). The adolescent brain and age-related behavioral manifestations. Neurosci Biobehav Rev, 24(4), 417-463. Retrieved from http://www.ncbi.nlm.nih.gov/pubmed/10817843

Wasserstein, R. L., Schirm, A. L., & Lazar, N. A. (2019). Moving to a World Beyond “p < 0.05”. The American Statistician, 73(sup1), 1-19. doi:10.1080/00031305.2019.1583913

Zhu, W. (2012). Sadly, the earth is still round (P < 0.05). Journal of sport and health science, 1, 9–11. doi:10.1016/j.jshs.2012.02.002

Zhu, W. (2016). p < 0.05, < 0.01, < 0.001, < 0.0001, < 0.00001, < 0.000001, or < 0.0000001 …. Journal of sport and health science, 5(1), 77-79. doi:10.1016/j.jshs.2016.01.019

Author Response

My co-authors and I are pleased to respond the reviewers’ comments and resubmit a revised version of the paper entitled: “Association between health-related physical fitness and self-rated risk of depression in adolescents: DADOS study” submitted for the special issue “Fitness, Physical Activity, and Health in Youth” at International Journal of Environmental Research and Public Health (No. IJERPH-778764). We have explained the points raised by the reviewers and we have modified the manuscript as requested. The changes made on the original manuscript appear in red text for ease of identification.

Yours sincerely,

The authors

REVIEWER 2

Abstract

Comment 1

“The association between fitness and depression in healthy adolescents is poorly understood”; I suggest to attenuating this statement, as there is already sufficient evidence of that kind of association.

Response

We thank the reviewer for the valuable comment. The suggested statement has been attenuated. Please, see line 13 and 14.

Comment 2

Report the gender-ratio.

Response

Done. Please, see line 16.

Comment 3

Spell-out DADOS, when mentioning the acronym for the first time.

Response

Done. Please, see lines 16 and 17.

Comment 4

I suggest to specifying the wording such as ‘self-rated symptoms of depression’, ‘objectively assessed fitness indices’, ‘higher cardiorespiratory fitness indices were inversely associated with a higher risk of self-rated symptoms of depression’, and similar.

Response

Following the reviewer’s comment, key terms have been properly used along the manuscript for an easy understanding of the readers. Authors have decided to use terms previously stablished in the scientific literature (i.e., self-rated risk of depression and objectively assessed physical fitness). Please, find these terms along the manuscript.

Comment 5

As regards the following beta-weight: β = -0.063; p < 0.05, please note that despite the ‘significant’ p-value, objectively assessed fitness indices were able to explain about 6% of the variance of self-rated symptoms of depression, or simply put: the predictive power of objectively assessed fitness indices is spurious.

Response

Thank you for your comment. Authors have realized about one mistake regarding beta values. This mistake has been properly amended in the abstract and in the results section. Please, see lines 21 and 22, and 162 to 164. In addition, following the reviewer’s comment, authors have tone down the discussion section according reviewer’s suggestion. Please, see lines 189, 191 and 212.

Introduction

Comment 6

Delete Beck et al 1961, as this reference describes a tool, but not diagnostic criteria. Rather cite the DSM-5 (American Psychiatric Association, 2013).

Response

Thank you. Done. See line 33.

Comment 7

“Probably due to a confluence of physical, hormonal, social, emotional and psychological changes”. Please add also neuronal changes (Giedd et al., 1999; Gogtay et al., 2004; Paus, Keshavan, & Giedd, 2008; Paus et al., 1999; Spear, 2000) and changes in sleep (Brand et al., 2014; Brand et al., 2017).

Response
Thank you for the valuable comment. Following the reviewer’s suggestion, neuronal and sleep changes have been mentioned in that part of the introduction section. Likewise, some of the proposed studies have been also included in the introduction section. Please, see lines 33 to 35.

Comment 8

Further, as regards stability and changes in physical activity indices, the authors should give a closer look at (Madsen, McCulloch, & Crawford, 2009); as regards the protective power of regular physical activity for symptoms of depression and parent-child issues, please give a close look at (Sigfusdottir, Asgeirsdottir, Sigurdsson, & Gudjonsson, 2011). For the emergence and maintenance of symptoms of depression in females, see (Bor, Dean, Najman, & Hayatbakhsh, 2014; Hyde, Mezulis, & Abramson, 2008). Next, the association between mental health and physical activity is not that linear and easy as suspected: (Opdal et al., 2019) nicely showed that changes and stability in psychological functioning and changes and stability in physical activity indices were completely unrelated. Next, the authors must give a very close look at (Bailey, Hetrick, Rosenbaum, Purcell, & Parker, 2017), when setting-up the theoretical background of their study.

Response

Thank you very much for the recommended studies. Authors agree with the reviewer in that physical activity may be relevant for health-related physical fitness and self-rated risk of depression in adolescents. However, since in our study, physical activity only acts as a covariate, after a deep discussion, authors have decided to reorganize the manuscript including this issue and the proposed references in the discussion section. Please, see line 199 to 204.

Comment 9

“Prior research has shown a strong positive association between weight status (overweight or obese adolescents) and depression [22,23]. Findings from reviews and meta-analyses studies have shown a positive relationship between weight status and depression [24]. In addittion (sic!), a systematic review and meta-analyses of longitudinal studies [7] suggested a bi-directional association between depression and obesity in adolescents.”; here, it is not clear, why this paragraph has been introduced; if weight is a dependent or an independent variable, then the title must be changed. Likewise, when weight is an issue, weight gain is a proxy of symptoms of eating disorders, and as such, the authors do not assess a “non-pathological” sample anymore.

Response

We agree with the reviewer that this is an issue that needs to be clarified in the manuscript. It is important to highlight that the theoretical frame of our study is based on the definition of health-related physical fitness stablished by Ruiz et al., (2009), which includes body composition within its definition, along with cardiorespiratory fitness, musculoskeletal and motor capacities. Following this suggestion, that paragraph has been deleted, and we have briefly mentioned prior findings about the relationship between body composition (as an independent variable) and depression in adolescents. Please, see lines 40 to 50.

Ruiz JR, Castro-Piñero J, Artero EG, et al. Predictive validity of health-related fitness in youth: a systematic review. Br J Sport Med. 2009; 43: 909-923.

Comment 10

Overall, the Introduction section needs a thorough revision; key concepts must be introduced and described in a much more detailed fashion; the state of-the-art must be thoroughly reported. In a similar vein, I strongly suggest to formulating hypotheses: based on the most recent findings (see above all see the more recent references mentioned above), what kind of pattern of results do they expect from their data and why? Likewise, do the present results add to the current literature in an important way, and if so, to what extent. Last, it remains puzzling, if among a sample of psychologically healthy adolescents (as stated by the authors) one could expect more than mere floor effects of self-rated symptoms of depression.

Response

According to the reviewer’s comments, the introduction section has been revised and reorganized, key concepts have been deeply described, and the hypothesis of the study has been clearly stated. Lastly, the term “healthy” has been removed from the whole manuscript to avoid confusion, since the study has been conducted with a general population of adolescents. Please, see lines 31, and 37 to 56.

Methods

Comment 11

How did the authors know or assess that participants were “free of any chronic disease”?. Here, the reader needs much more information. Relatedly, does “chronic disease” relate exclusively to physical/somatic issues such as diabetes, obesity, sleep-disordered breathing such as Obstructive Sleep Apnea, neurological issues such as early-onset of multiple sclerosis, asthma, celiac disease, or irritable bowel disease, just to name a few, or does “chronic disease” also relate to psychiatric diseases such as ADHD (prevalence rate: 5.6% (C. A. Polanczyk, de Lima, Horta, Biederman, & Rohde, 2007; C. A. Polanczyk, Willcutt, Salum, Kieling, & Rohde, 2014; G.V. Polanczyk, Salum, Sugaya, Caye, & Rohde, 2015), or internalizing problems (Bor et al., 2014)?

Response

Thank you for the question. Following the reviewer’s comment, we specify that the chronic diseases mentioned referred to physical or neurological diseases. We also mention that it was self-reported by participants’ parents. Please, see lines 62 to 65.

Comment 12

The Declaration of Helsinki was first stated in 1964, not 1961.

Response

Done. See line 71.

Comment 13

Risk of depression; please specify, who completed which part of the Behaviour Assessment System for Children.

Response

Thank you. The Behaviour Assessment System for Children was completed by the adolescents. Authors have clarified this information in the methods section. See lines 117 to 122.

Comment 14

Statistical analysis: I strongly recommend to report effect sizes; this holds particularly true, as the sample is quite large; it follows that due to the sample size the p-value might get ‘significant’, while mean differences are still spurious and trivial.

Response

Thank you for the valuable comment. According to the reviewer’s suggestion, the effect size has been reported in all the analyses. Cohen’s d has been included in Table 1 as measure of effect size for mean differences. R2 (Table 2) and R2 of Cox/Snell (Table 3) have been reported for regression analyses. Additional information has been included in the methods section and in tables. Please, see lines 142, 143, 150 and 151.

Cox, D. R., & Snell, E. J. (1989). The Analysis of Binary data (2nd ed.). London: Chapman and HAll.

Comment 15

Results: Table 1; please report all statistical indices; reporting just p-values does not make sense (Wasserstein, Schirm, & Lazar, 2019; Zhu, 2012, 2016).

Response

According to the reviewer’s comments, and as previously mentioned, Cohen’s d has been included in Table 1.

Discussion

Comment 16

Discussion: as for the Introduction section, the Discussion section needs a thorough revision. Among others and specifically, the authors should critically comment on the low beta-weights.

Response

We thank the reviewer very much for the suggestion. We have reformulated and expanded the discussion section. Please, see lines 186 to 240.

Round 2

Reviewer 2 Report

Please see the detailed comments as stickers in the pdf-file. 
